# Fibrous Materials for Potential Efficient Energy Recovery at Low-Temperature Heat

**Patrizia Frontera** [1,2,*] , **Lucio Bonaccorsi** [1,2] , **Antonio Fotia** [1,2] and **Angela Malara** [1,2,*]

1   Department of Civil, Energy, Environmental and Material Engineering, Mediterranea University of Reggio Calabria, 89124 Reggio Calabria, Italy
2   National Interuniversity Consortium of Materials Science and Technology (INSTM), 50121 Florence, Italy
*   Correspondence: patrizia.frontera@unirc.it (P.F.); angela.malara@unirc.it (A.M.)

**Abstract:** Technology must improve energy generation and utilization to support human societies. All highly industrialized nations support the attempt to switch from fossil fuels to renewable energy sources—a process which is irreversible—but the support is not yet strong enough to make the switch. Energy-efficient and renewable heating and cooling systems offer considerable energy saving potential, since buildings use a large percentage of EU energy for heating and cooling, which still uses fossil fuels (75%). For this transition, innovation regarding the traditional material for thermal energy storage appears to be crucial. This work proposes a review of a new approach to thermochemical materials for energy recovery in the low-temperature range, based on the production of microfibers by electrospinning. The novelty of applying fibrous materials in thermal energy storage systems is related to the particular configuration of the adsorbing phase and the production technique used. Microfibers show a large surface area, high vapor permeability, and high structural stability, and they can be easily electrospun to form self-standing foils or coatings for heat exchangers.

**Keywords:** fibers; electrospinning; adsorbent materials; energy recovery; low-temperature heat

## 1. Introduction

The efficient production and use of energy is a process of technological development essential for the sustainable growth of human societies. The effort now supported by all highly industrialized countries to convert the production of energy from traditional fossil fuels to sources with a low environmental impact, for example, by increasing production from renewable sources, is an irreversible process that has now started, but is not yet sufficient to allow for the transition. The environmental impact of the current technologies for the production and use of energy, the climate emergency, and the geopolitical tensions surrounding the concentrated distribution of fossil fuels identify the need to enhance alternative forms of energy production (such as renewable energy). On the other hand, however, they require better efficiency in energy consumption by reducing waste and environmental dispersion. Indeed, due to the consumption of a big share of EU energy for the heating and cooling of buildings, which still relies on fossil fuels as its dominant energy source (75%) [1], energy-efficient and renewable-based heating and cooling systems have high potential for energy savings. In its vision for 2050, the Committee on Climate Change [2] projects that fossil fuels will be replaced by heat pumps (52%), low-carbon heat networks in heat-dense areas (such as cities) (42%), hydrogen boilers (potentially) (5%), and direct electric heating (1%).

According to recent studies [3], about 70% of the world's energy is dissipated during its use, and about 50% of this is lost in the form of thermal energy by vapors, hot gases, and heat carried by water and air-cooling systems. It is estimated that in the countries of the European Union 70% of the energy used in the industrial sector is expended for thermal processes (ovens, reactors, boilers, dryers, etc.), and up to one-third of this energy is lost [4].

This is an energy potential of around 300 TWh/year that could be partially recovered by increasing the efficiency of energy use, while simultaneously reducing harmful emissions.

The difficulty in recovering this enormous energy potential lies in the fact that only 15% of all the wasted thermal energy is at high temperature, that is, in a temperature range between 500 and 1000 °C, approximately the 25% is at medium temperature, in the range between 200–500 °C, and the residual amount of about 60% is the so-called low temperature energy, at temperatures below 150 °C. From a technological point of view, the recovery of thermal energy is more easily achievable when the temperature of the dissipated flows is in a medium-high range, thanks to the availability of a high temperature gradient, while it is much more difficult at low temperatures [5].

Three methods are commonly used to store thermal energy: as sensible heat, latent heat, and reaction heat (thermochemical storage, TES), as schematically depicted in Figure 1. In the first case, the temperature of the materials with high thermal capacity is raised. This is an economic system with a low storage density (in the order of 50 kWh/m$^3$) that requires high temperature gradients and is not suitable for long-term storage. In the second case, the materials undergo phase transitions, such as solid/liquid, and have an accumulation density (100 kWh/m$^3$) higher than the former. Again, these materials are unsuitable for long-term storage and have low thermal conductivity, which reduces the efficiency of energy recovery. Finally, thermochemical energy storage is based on the properties exhibited by some materials of exchanging thermal energy when they are in contact with suitable fluids because of chemical-physical transformations that involve the accumulation of thermal energy (endothermic transformation) and its release (exothermic transformation) in a reversible, economical way, even over long periods of time, with minimal losses and high energy density (500 kWh/m$^3$) [6]. In general, TES is based on a solid material that reacts with water as a thermal fluid because this coupling guarantees the absence of harmful components, zero environmental impact, and the cost-effectiveness of the system.

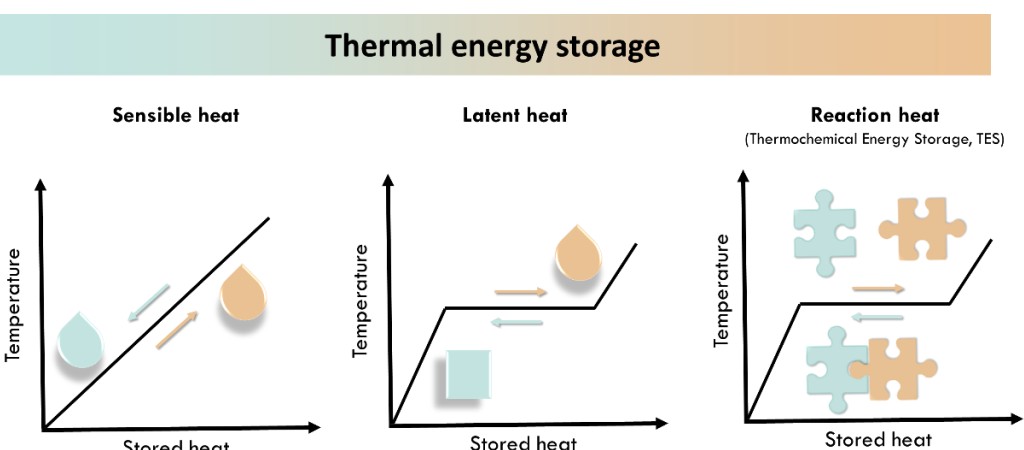

**Figure 1.** Schematic depiction of thermal energy storage methods: sensible heat, latent heat, and reaction heat.

A TES system consists of a reactor in which the solid material is in contact with a heat exchanger which, during the storage phase (endothermic/activation phase), heats the material by means of a wasted thermal flow (exhaust fumes, hot water, hot air, etc.), causing the desorption of water molecules from its structure. The evaporated water is condensed and stored in an auxiliary tank, in the case of closed cycles, or released into the atmosphere (open cycles). The material can be kept dehydrated in the reactor for medium (seasonal storage) or even long periods of time. In the thermal energy recovery phase (exothermic/deactivation phase), the material is rehydrated with the condensed water contained in the auxiliary tank (closed cycle) or with the humidity of the atmosphere (open cycle) and releases the hydration heat to the heat exchanger, which will then transfer it to the user. TES appliances are therefore easily buildable, do not contain harmful

components, are relatively cheap, and are suitable for use in smart buildings in industrial and civil environments.

The major drawback of TES systems is the high thermal resistance at the interface between the solid material (in the form of granules) and the heat exchanger in the reactor that reduces the heat transfer efficiency [7]. The contact area between granules and metal surfaces of the heat exchanger is, indeed, limited, and the heat transfer during the storage and recovery phases occurs at the granules/metal interface by conduction. Finally, the storage materials, exposed to multiple hydration/dehydration cycles, must ensure that no morphological alterations of their structure occur to avoid a reduction in the surface area and thus, a decrease in the performance or decomposition, or the formation of corrosive products over time [8]. Many of the thermochemical materials used today do not exhibit the necessary structural stability.

For all the above reasons, in this work, the literature was revised and summarized, and the newest approach for the use of thermochemical materials for energy recovery in the low-temperature range was proposed, based on the production of microfibers by electrospinning. The novelty of applying fibrous materials in TES systems is related to the particular configuration of the adsorbing phase and the production technique used. Microfibers show a large surface area, high vapor permeability, and high structural stability, and they can be easily electrospun to form self-standing foils or coatings for heat exchangers.

## 2. Adsorbent Fibrous Material Synthesis and Characterization

Adsorbent materials are usually commercialized in a variety of granules and powders and are typically used in granular form in adsorption systems to fill the gap between the fins of a heat exchanger in order to produce the heat pump's adsorber module [9]. However, this design has significant limitations for two primary reasons: (i) at the adsorbent material/metal interface, heat transfer is highly resistant due to the small contact area between the granules and the fins; (ii) a heat exchanger's appropriate integration and containment of loose adsorbent material grains provide certain practical challenges and are barely scalable at an industrial level [10].

Thus, researchers are devoting their efforts to developing several adsorbent coatings for heat exchangers to overcome heat transfer and density limitations, such as loose grains, of standard adsorbent bed configurations [11]. Among the attempted approaches is the development of composite structures combining the adsorbent materials with metal foams and fibers, which have been utilized to reduce mass and heat transport limitations in adsorption heat exchangers [12,13]. Wittstadt et al. suggested that the limitations of diffusion in the adsorbent pores can be overcome by a structure made up of thin fibers leading to thinner adsorbent layers with the same mass ratio and allowing higher power densities [14,15]. Further, the possibility of producing multi-layered fibrous materials with middle layered structures potentially allows for the achievement of modulable thermal behavior [16].

Fibrous morphologies with nanoscale dimensions provide a plethora of intriguing characteristics, including a superior mechanical performance and a high surface-to-volume ratio, which makes fibers desirable for a variety of applications [17]. In addition to their vast surface area, they also have a high functionalization capacity. All these aspects contributed to suggest that the fibrous morphology is particularly promising for the development of coatings for heat pump adsorbers (Figure 2). Despite the fact that manifold techniques can be used to develop fibrous materials, electrospinning is a rapidly emerging as a simple method to produce highly porous structures of smooth, nonwoven nanofibers. It basically uses electric force to pull yarns of solutions or molten polymers into diameters in the order of hundreds of nanometers. It is notable that the investments of numerous companies in electrospinning technology have increased the annual production capacity of nanofibers to the scale of metric tons [18,19]. Indeed, it is considered a very simple, scalable and inexpensive technique for the production of one-dimensional nanostructures with different morphologies, such as hollow, dense and core-shell fibers, that can also have an organic,

inorganic, and hybrid composition [20]. In addition, a noteworthy advantage in the electrospinning technique is the possibility of depositing the nanofiber coating only in the required zones, which may aid in the minimization of material wastage.

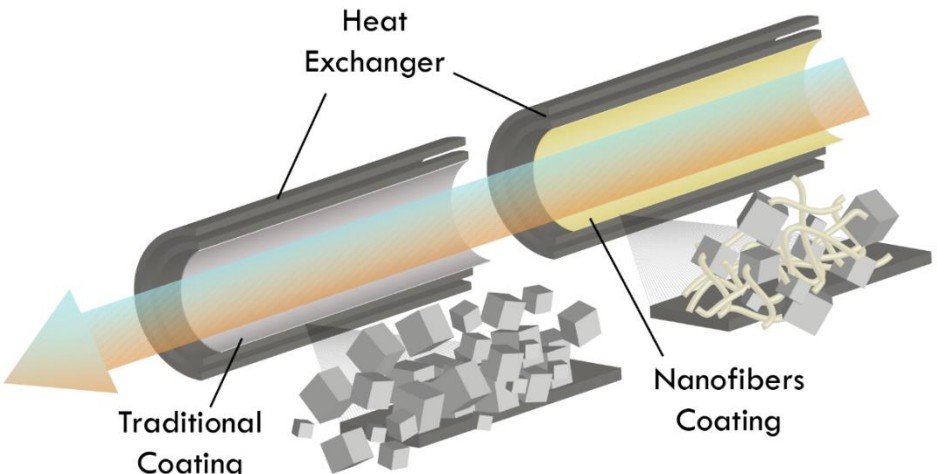

**Figure 2.** The configuration of powder and fibrous adsorbent materials deposited in the heat exchanger apparatus.

Electrospinning shares characteristics of both electrospraying and traditional dry fiber spinning. However, the process does not require the use of chemical coagulation or high temperatures to produce fibers from solutions [21]. This makes the method particularly suitable for the production of fibers using large or complex molecules. Electrospinning can also be performed from molten precursors; this method guarantees that solvents will not be transferred to the final product.

Electrospinning works similarly to the principle of an inkjet printer, where droplets of polymer solution are projected onto a surface (Figure 3). A high voltage is applied to the polymer droplet, which then charges with electricity. When the electrostatic repulsion is greater than the surface tension, the drop is deformed into a conical-shaped structure known as a Taylor cone. The jet of liquid is then ejected towards the collector, and during the flight, it evaporates and lengthens, forming a continuous filament which is finally deposited randomly on the collector surface, as solid micro- or nanofibers. A non-woven and ultra-porous layer is then formed. Many parameters influence the properties of the layer obtained: the choice of polymers, the concentration and the viscosity of the solution, the electrical potential, the distance between the capillary and the collection surface, and the ambient parameters (temperature, humidity, etc.) [22]. Moreover, electrospinning is a process that creates nanoscale fibers by using an electric field to pull and stretch a polymer solution or melt. The resulting fibers have a high surface area-to-volume ratio, which makes them useful for a variety of applications (tissue engineering, energy storage, sensing, drug delivery) [23]. Further, fibers can be functionalized with different kinds of molecules, allowing the possibility of interaction with specific analytes [24].

It has been extensively discussed that the fibrous form of materials can generally enhance all their main featuried properties. This is still valid when composite adsorbent/fibrous materials are considered. Indeed, adsorbent properties can be improved or better promoted when their spatial configuration is optimized, as in the case of the so called "necklace" morphology, where the adsorbent crystals are regularly distributed along filaments of the fibrous matrix [20]. In particular, experimental results of previous studies demonstrated that hybrid microfibers maintained their adsorption properties to the water vapor of powder adsorbents, were thermally stable, and had a morphology particularly apt to create adsorptive coatings on the heat exchanger surface [25]. As will be widely discussed in the following sections for the specific fibrous/adsorbent couples, alongside the textural and mechanical features, the most characteristic properties in fibrous adsorption

systems are the water vapor adsorption/desorption capability, the total surface area, and the thermal stability.

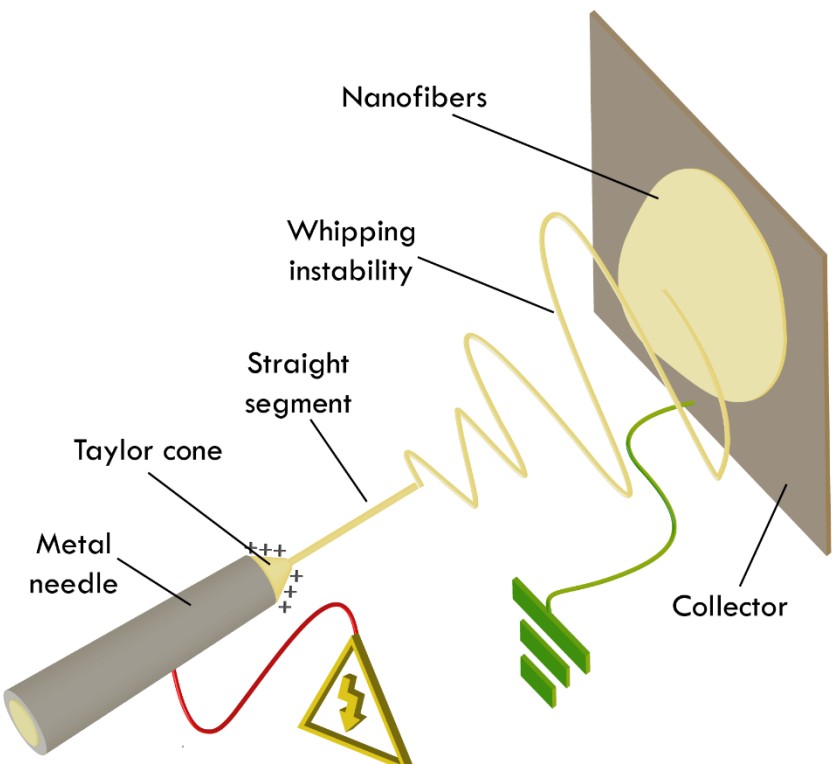

**Figure 3.** Schematic electrospinning setup.

Textural properties are generally investigated by means of electron microscopy and are strictly related to morphology aspects, which, as expected, are dependent on several specifications due to the production conditions. Polymer based fibers are one-dimensional structures characterized by diameters ranging between tens of nanometers to a few microns, which are generally solid, homogeneous, and regular. Normally, micro and nanofibers produced by electrospinning have different configurations in terms of orientation and hierarchical structures, as well as in terms of their arrangement, which can be either random or aligned [26].

The mechanical properties of electrospun fibers, such as tensile modulus and tensile strength, are strictly dependent on the morphological structure of the fibers [19]. Indeed, the great variation in mechanical properties upon considering the single fiber or the entire mat has been extensively discussed. Moreover, the fiber chemistry (such the composition of the solution), as well as the electrospinning parameters (process, solution, and environmental) and post-treatment processing greatly influence the uniformity of fibers and their diameter. Despite the fact that micro- and nano-tensile testing systems can be used to measure the modulus of a single fiber to optimize the electrospinning conditions, it is generally preferred to tailor the mechanical properties of the final electrospun product from the data obtained for a mat, since it is easier to test a fibrous mat compared to individual fibers [27]. However, due to the porosity of fibrous systems, together with the random orientation of fibers, sometimes it is difficult interpret the bulk mechanical properties, since they can be very different among the same sampled materials, being related to the variable microscopic and macroscopic characteristics of the electrospun mats. Clearly, when hybrid composite materials are considered, the presence of different typologies of additives can alter the mechanical behavior, and each case must be analyzed with specific considerations. It is therefore evident that mechanical stability is an important issue in terms of stiffness and resistance for coatings applications.

Large specific surface area and inter-fiber porosity are also important parameters that are generally useful when material adsorption properties are considered. The porous structure is beneficial to the increase in the specific surface area of fibers and plays a pivotal role in the permeability characteristic. Moreover, the net shape configuration may also contribute in the possibility of physically entrapping the additive particles in the mesh of the fibrous structure, in addition to the previously claimed necklace morphology and particle encapsulation. A schematic illustration of the possible relative arrangement between the fibers and the additive is displayed in Figure 4. Those systems, as will be discussed in the following sections, are considered highly innovative, since they are capable of blocking particles of very different dimensions in a polymeric permeable structure without impairing their properties, such as adsorption, in contrast with what happens in traditional systems, where additive features can be partially inhibited. In this context water vapor adsorption/desorption ability, generally ascribed to additives, is always guaranteed.

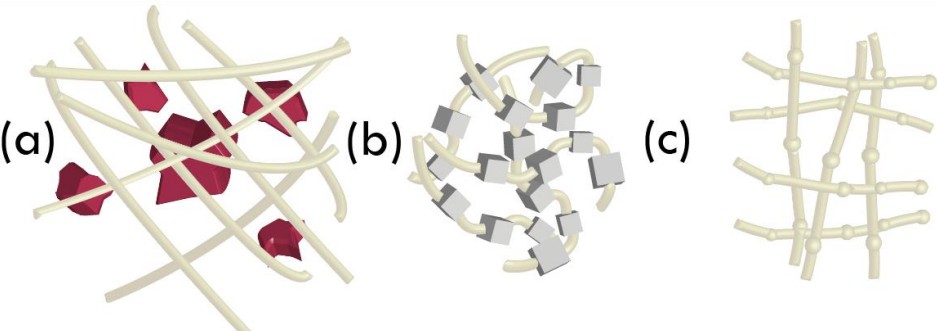

**Figure 4.** Encapsulation arrangement between fibers and additives: (**a**) irregular particles entrapped in the fibrous matrix, (**b**) regular particles bound in the so called "necklace" morphology, and (**c**) round-shaped particles embedded in the polymeric fibers.

A critical aspect is instead related to the thermal stability of fibrous systems, considering that the polymeric matrix developed with the electrospinning apparatus is obviously subjected to the chemical/physical characteristics of the polymeric phase. Indeed, electrospun fibers are effectively thermally stable in a reduced temperature range. However, it is important to highlight the very low operation temperatures considered in the abovementioned applications, which are fully compatible with the electrospun mats, and are therefore sufficient to avoid the degradation process. Polymer-based fibers were also reported to be stable and to maintain their structure without evidence of swelling or thermal damage, even after a number of adsorption/desorption cycles [7].

### 3. Hybrid Silica Fibers

Silica gel is one of the most widely utilized porous materials in adsorption chillers due to its low price and wide market availability. Silicon alkoxides, or organic salts, are used to produce porous silica gels, which are then progressively subjected to hydrolysis and condensation, aging, drying, and calcination [28]. On the surface of porous silica, there are both silanol (Si-OH) and siloxane groups (Si-O-Si). Silanols are considered strong adsorption sites, whereas siloxanes are hydrophobic sites. Silanols may be solitary, vicinal, or geminate and may be connected to the surface water via hydrogen bonds. The significance of hydroxyl groups from an adsorption standpoint stems from the fact that the adsorption process is more efficient the greater the proportion of active silanols per unit of surface area.

The moderate heating of silica in a vacuum (100–120 °C) ensures the nearly total elimination of physically adsorbed water. However, the last water monolayer is removed at an activation temperature of 200 °C, whereas the use of higher temperatures (200–1000 °C) facilitates the removal of chemisorbed water. Despite the numerous synthesis techniques to produce various silica nanoparticles and nanocomposites, the process of synthesis should

be focused on the improvement of adsorption sites and their stability for use as adsorbent materials in engineering applications.

Silica gel is typically utilized in granular form in adsorption systems to fill the area between the fins of a heat exchanger to create the adsorber module of heat pumps [29].

In order to obtain fibers of silica by electrospinning, two approaches are suitable. The first is the combination of electrospinning/sol–gel processes that allows for achieving the formation of silica nanofibers by using precursors of a synthesis solution with suitable viscosity to be electrospun. Generally speaking, there has been a growing interest in the fabrication of nanofibers of binary metal oxides by combining the sol–gel method and the electrospinning process [30]. Following this method, starting from solutions having TEOS/ethanol/water/HCl = 1/2/2/0.01 in a molar ratio and aged at 80 °C for 30 min, electrospun silica fibers at two different applied voltages, 10 and 12 kV, were obtained [31]. The addition of the acid in the solution synthesis promotes the formation of linear structures, and the solution was ripened to increase the solution viscosity and to further enhance the linear structure. Finally, in order to obtain the silica fibrous structure, fibers were calcinated to decompose the polymeric carrier X. However, the obtained pure silica microfibers show a mechanical behavior that is considered too brittle to be used as a coating for adsorption heat transformer applications, as previously discussed. Therefore, Frontera et al. [32] recently proposed a hybrid fibrous coating in which the organic–inorganic nature of microfibers was preserved. Indeed, the polymeric phase made of polyvinylpyrrolidone (PVP) was maintained and deeply bonded to tetraethyl orthosilicate (TEOS) to ensure thermal stability, structural flexibility, and long-term usability, without compromising the adsorbent properties. In particular, the FTIR spectroscopic analysis proved that, in well-defined conditions of PVP/TEOS percentage ratio, TEOS interacts with PVP, interfering with the solvent evaporation during the electrospinning process, and with the final capacity of the material to adsorb water. Moreover, the formation of silica particles enwrapped by PVP molecules before their further aggregation into larger clusters is favored by ES, while at higher percentages, TEOS yield bigger, water-rich silica structures interacting with PVP [33]. It was finally discussed that silicate addition reduced the original water affinity of PVP and in detail, a high amount of siloxanes bonded to the surface of the microfibers, delaying the thermal degradation of the polymeric component, whereas hybrid microfibers with low TEOS concentration demonstrated adsorption properties similar to those of commercial silica gel and good short term thermal stability [32,33]. The mechanical strength was preserved by maintaining the hybrid, organic–inorganic structure of microfibers.

The second approach fabricates the hybrid fiber oxide with blended polymer solution to produce suitable electrospinning conditions through intermixing and interaction between the silica oxide and selected polymers. Depending on the fibers, application of the thermal treatment is not mandatory for the removing polymer. The hybrid fibers with respect to only the silica fibers exhibit different enhancements in terms of: (i) mechanical properties; (ii) adhesion to the substrate; and (iii) limited loose materials after the adsorption-desorption cycles.

Following this approach, a new hybrid coating made of microfibers obtained by the electrospinning of silica gel/polymer solutions was produced by Freni et al. [25]. Polyacrylonitrile (20 wt%) and different silica gel powders (80 wt%) were mixed and electrospun to produce fibrous coatings on aluminum surfaces (Figure 5). The performances of the coatings in terms of water adsorption obtained depends on the source of silica gel; indeed, the polymeric phase in the microfibers coatings did not obstruct the silica powder porosity, preserving the water adsorption ability (Figure 6). The hybrid coatings produced by electrospinning were morphologically and functionally stable in the temperature range of the heat pump cycle. Indeed, the thermal stability of hybrid coatings was studied by thermogravimetry up to 500 °C in order to investigate the degradation trend of the polymeric component, although the operating temperature range of silica gel adsorption chillers is limited to the range 100–130 °C [25]. Due to the dispersion of the silica granules, the permeable structure of the electrospun coatings has shown some advantages, such as

the flexibility to directly coat the metallic surfaces of the heat pump adsorber, as well as the production of self-supporting fabric of hybrid microfibers with intrinsic mechanical resistance [25]. Moreover, using a different polymer as a polymethylmethacrylate, smaller silica particles are embedded or glued in the polymers fibers, obtaining a maximum water uptake of the textile comparable to the silica gel concentration in the hybrid coating, but with a detrimental effect on the mechanical response [27].

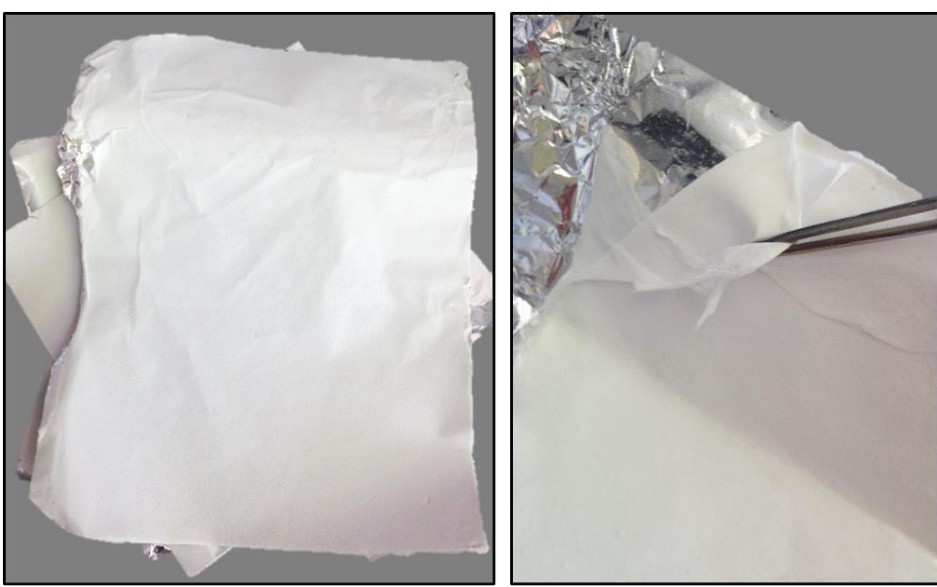

**Figure 5.** Electrospun PAN/silica gel flat coating with a paper-like appearance.

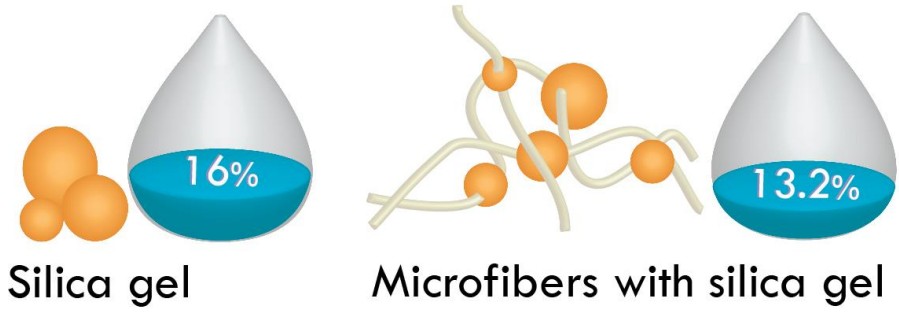

**Figure 6.** Water adsorption ability of silica gel powder and PAN/silica gel fiber coating.

## 4. Hybrid Microporous Materials—Fibers

The advances in materials sciences have substantially promoted the development of micro adsorbents, and within this related material, zeolites are extensively tested for use in adsorption heat pumps [34]. Zeolites have a higher water affinity than other materials because of their electrostatic charge, which promotes the adsorption of polar molecules. Additionally, a mathematical model shows that zeolite-coated adsorbers for adsorption pumps perform better than alternative configurations in terms of specific and volumetric powers [35]. However, the aluminosilicate zeolites have the drawback of requiring high regeneration temperatures (and high thermal energy expenditure), restricting the use of these materials to systems where a high temperature-driving heat source (150–300 °C) is available. The materials that overcome this limitation are a new class of zeolite materials, denominated AlPO and SAPO, able to combine a moderate hydrophilicity with a high capacity of water vapor adsorption, resulting in a fairly low regeneration temperature (60–100 °C) and a lowered desorption heat, maintaining excellent performance.

SAPO-34 cube-rounded crystal coatings were deposited in situ on different metallic supports, highlighting that the adhesion of zeolite coatings on stainless steel and copper

was poor. On aluminum supports, a better adhesion was observed, which was explained by the aluminum oxi/hydroxide layers' active development in the formation of chemical interactions at the zeolite–metal interface [36].

Different ex situ coating techniques (dip-coating, drop casting, spray coating) consisting of the bonding (indirect) thermal contact between the adsorbent and the heat conductive metal are proposed [37]. SAPO-34 zeolite powder was added in a sulfonate polyether ether ketone(S-PEEK) binder at four different percentages (from 80 wt% up to 95 wt%), obtaining a coating homogeneous and compact, with effective water sorption permeability [38].

Recently, hybrid SAPO-34 microfibers obtained by electrospinning for water adsorption based on PAN (polyacrylonitrile), PVA (polyvinyl acetate), PEO (polyethylene oxide), PS (polystyrene), and PMMA (polymethyl methacrylate) as the polymeric component, were specifically produced as coatings for adsorption systems driven by low temperature heat sources [20,27,39,40]. The suitability of the electrospinning technique to obtain the uniform and homogeneous coating, despite the different combined phases (solid for SAPO-34, liquid for polymer and solvent), is highlighted by the coating deposition on prototypal scale, as in the case of the SAPO-34/PAN pair (Figure 7). The aim was to obtain the highest concentration of SAPO-34 in the coating, as high as 85 wt%, while maintaining the good structural and mechanical properties of the microfibers (Figure 8). The polymer mats preserve the adsorbent filler's abilities by not interfering with the adsorption process; in fact, the polymer does not obstruct the water vapor movement, and the zeolite adsorption effectiveness is maintained in all the analyzed cases [20,27,40]. Additionally, the hybrid fibers demonstrated a maximum water vapor uptake that was equivalent to that of the traditional adsorbent materials often used in adsorption heat pump systems [41], even when different SAPO-34 zeolite sizes where considered [40]. The hybrid coating exhibits high thermal stability; thus, it can operate in the typical conditions of adsorption chillers driven by waste heat at temperatures of up to 150 °C. Moreover, from a mechanical point of view, a noteworthy improvement in the mechanical properties of microfiber coatings was reported. Indeed, the fibrous nature of the fracture surface produced in the mat suggests a nonbrittle progression of the fracture with respect to the traditional brittle and defective zeolite crystal coatings [20].

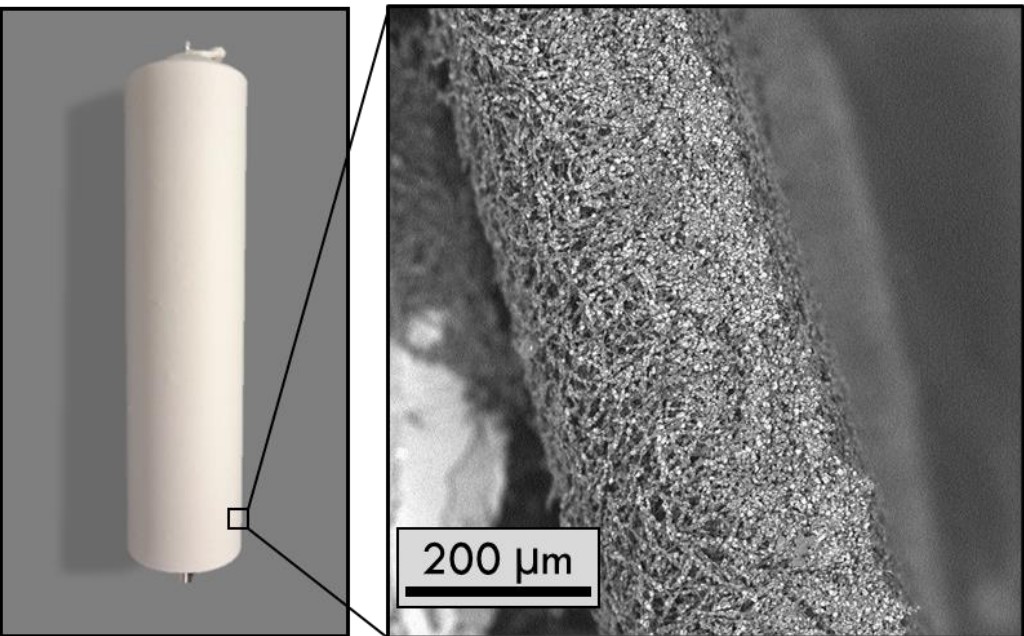

**Figure 7.** Electrospun PAN/SAPO-34 tubular coating and relative SEM micrograph.

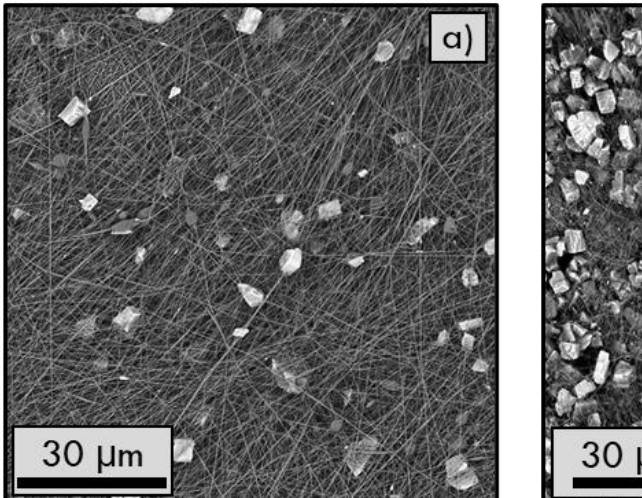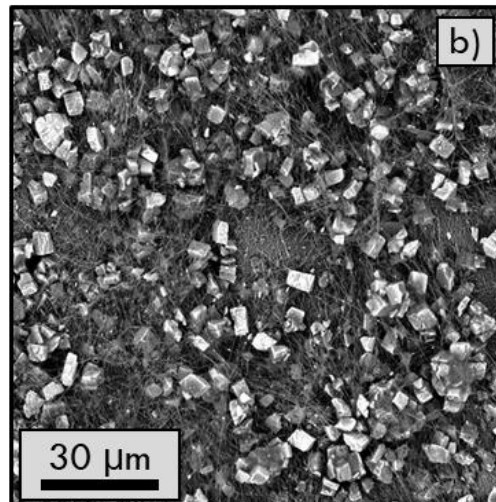

**Figure 8.** SEM micrographs of electrospun PAN/SAPO-34: (**a**) low SAPO-34 concentration, and (**b**) high SAPO-34 concentration.

## 5. Enhancement of Hybrid Fibers

Among the strategies to enhance the performance of TES storage materials, the use of hydrated salts have been recently reconsidered for thermochemical storage at low temperatures, below 100 °C, being easily dehydratable and with a good energy density [42]. The major limitation of hydrated salts is their poor structural stability, which can give rise to deliquescence or fusion in the most hydrated states, even at low temperatures. For these reasons, hydrated salts are difficult to use as storage materials, despite the numerous advantages they present. Therefore, the latest research in this field is currently focused on the possibility of synthesizing and characterizing hybrid microfibers obtained by the electrospinning of hydrated salts opportunely encapsulating along the fibrous structure, which could potentially show high structural stability and high resistance to the deliquescence phenomena. Such a characteristic could therefore be particularly indicated in the thermochemical storage field and particularly usable in heat pumping applications.

Preliminary results confirmed these purposes; in fact, polymer-based microfibers, with calcium chloride as the hydrated salt, were realized, starting with a polyvinylpyrrolidone (PVP) precursor and a hydrophilic polymer, solubilized in ethanol. Once the electrospinning parameters were set, hybrid fibrous mats with a paper-like appearance were produced (Figure 9); a uniform distribution of fibers with homogeneous diameters is evident both in PVP and hybrid PVP/salt fibers. Surprisingly, no evidence of calcium chloride crystals emerged from SEM analysis, but EDX proved the presence of calcium and chlorine (Figure 9). Likely, Ca and Cl ions are strictly connected with the PVP polymeric unit to form a homogeneous hybrid fiber. Thermogravimetry and DSC analyses were used to study thermal stability. In particular, pure precursors and salt exhibit water desorption at low temperatures, as demonstrated by thermograms (Figure 10). The heat degradation of the polymer began once the polymeric mats reached a stability limit of 300 °C. Adsorption curves (not shown) were measured by a volumetric system in the typical range of pressure and temperature used in low T adsorption/desorption cycles, such as 10 mbar. The water uptake is barely 10% for pure PVP, considering that it is a hydrophilic polymer, and almost 40% for the calcium chloride. Then $CaCl_2$ addition improves PVP microfiber adsorption, with no evidence of property degradation after 5 cycles of adsorption/desorption. These preliminary results highlight that hybrid microfibers of hydrated salts realized by the electrospinning technique can be addressed for application in low temperature heat storage.

All the analyzed systems and their related characteristics in terms of adsorption properties, as well as thermal and mechanical stability, are reported in Table 1.

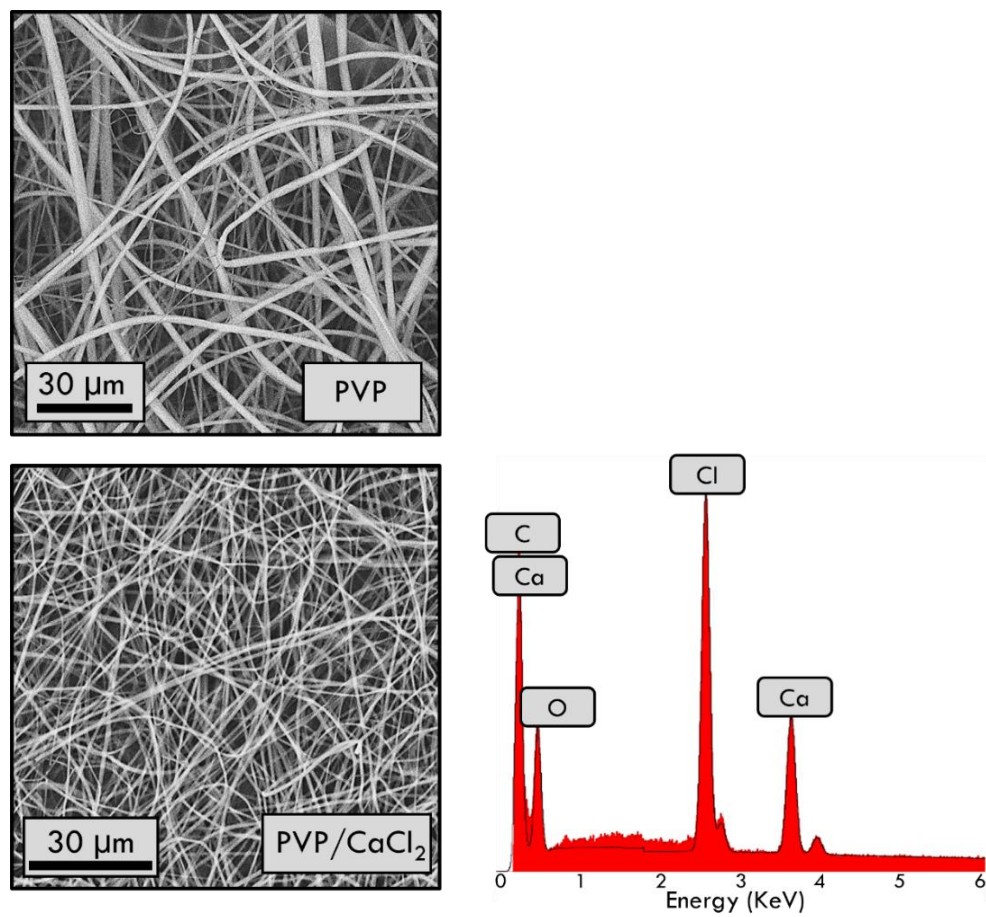

**Figure 9.** SEM micrograph of PVP fibers and SEM-EDX analysis of electrospun PVP/CaCl$_2$ coating.

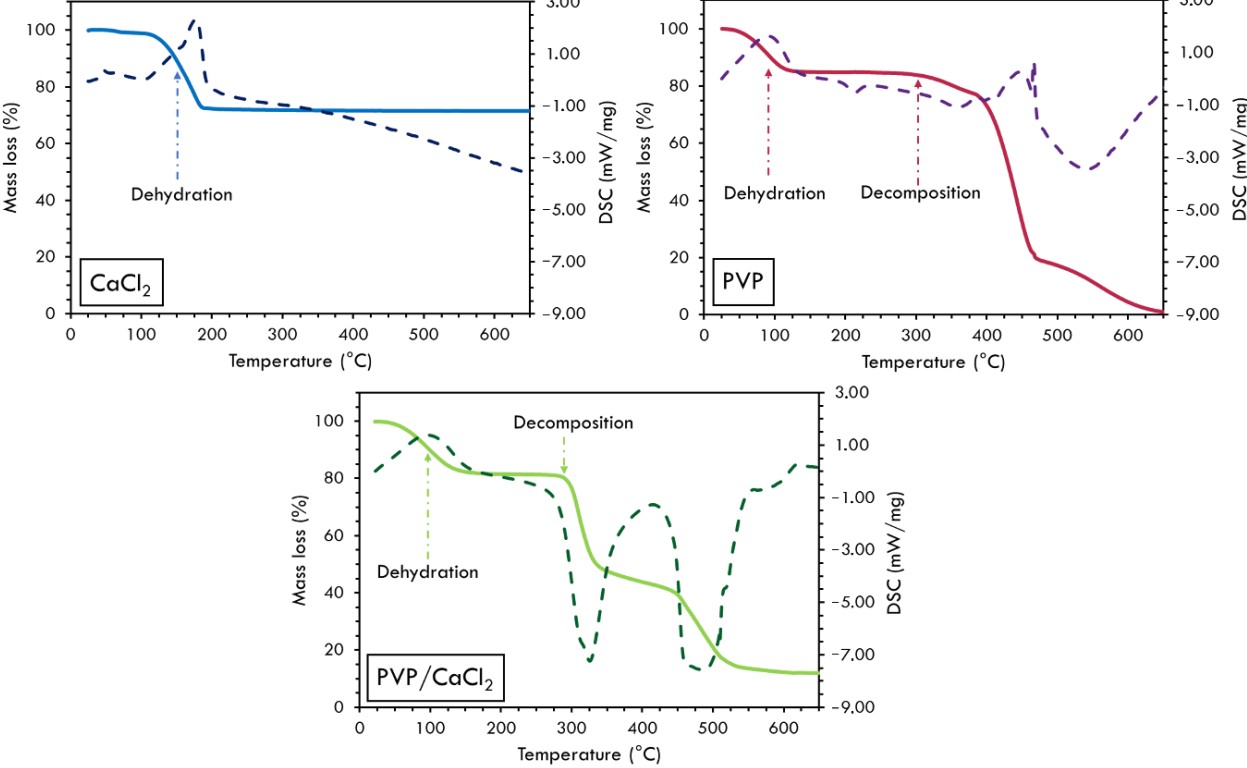

**Figure 10.** Thermograms of CaCl$_2$ salt, pure PVP fibers, and PVP/CaCl$_2$ fibers.

**Table 1.** Summary of the main analyzed systems and their related characteristics.

| Sample | Polymer/Support | Adsorbent Type | Production Technology | Thermal Stability (@ The Operating T) | Adsorption Properties | Mechanical Properties | Ref. |
|---|---|---|---|---|---|---|---|
| $SiO_2/TiO_2$ | - | TEOS | Electrospinning | √ | - | - | [30] |
| PVP/TEOS | PVP | TEOS | Electrospinning | √ | √ | √ | [25,33] |
| PAN/Si gel | PAN | Silica gel | Electrospinning | √ | √ | √ | [25] |
| PMMA/Si gel | PMMA | Silica gel | Electrospinning | √ | √ | × | [27] |
| SAPO-34-SPEEK | S-PEEK | SAPO-34 | Drop casting | √ | √ | √ | [38] |
| PAN/Zeolite | PAN | SAPO-34 (various grains size) | Electrospinning | √ | √ | √ | [40] |
| PMMA/Zeolite | PMMA | SAPO-34 | Electrospinning | √ | √ | √ | [27] |
| PVA/zeolite | PVA | SAPO-34 | Electrospinning | √ | √ | √ | [20] |
| PEO/Zeolite | PEO | SAPO-34 | Electrospinning | × | × | √ | [20] |
| PS/Zeolite | PS | SAPO-34 | Electrospinning | √ | × | √ | [20] |
| Hydrated salts | Porous matrix | Inorganic salts | Impregnation, encapsulation | × | √ | × | [42] |
| $PVP/CaCl_2$ | PVP | $CaCl_2$ | Electrospinning | √ | √ | √ | This study |

## 6. Conclusions

Over the last decade, advances in material technology have been sought to favor the transition to a low-carbon economy. Among the new approaches, electrospinning techniques have led to the structure of nanofibrous arrangements evolving from a nonwoven form to yarn, 3D assemblies, and patterned structures. This review covers a specific segment of thermochemical materials used in heat pumps produced in the form of microfibers by electrospinning, with the goal of energy recovery in the low-temperature region. The utilization of fibrous materials in thermal energy storage systems is an innovative concept that can be traced back to the specific configuration of the adsorbing phase, as well as the manufacturing method that is implemented. Microfibers have a huge surface area, high vapor permeability, and high structural stability, and may easily be electrospun into self-standing foils or coatings for heat exchangers. More complex structures can be created using electrospinning techniques, and determining the underlying concepts of how these techniques might possibly be paired with one another to provide more control over the process is an area for future research. Electrospinning will be adopted more widely in industry because electrospinning has a relatively low cost, requires a low quantity of raw materials, is simple to maintain, and makes it easy to fabricate nanofibers.

**Author Contributions:** Conceptualization, P.F., L.B., A.F. and A.M.; methodology, P.F. and L.B.; formal analysis, P.F., L.B., A.F. and A.M.; investigation, P.F., L.B., A.F. and A.M.; data curation, A.F. and A.M.; writing—original draft preparation, P.F., L.B., A.F. and A.M.; writing—review and editing, P.F., A.F. and A.M.; supervision, P.F. and A.M. All authors have read and agreed to the published version of the manuscript.

**Funding:** This research received no external funding.

**Institutional Review Board Statement:** Not applicable.

**Informed Consent Statement:** Not applicable.

**Data Availability Statement:** Not applicable.

**Conflicts of Interest:** The authors declare no conflict of interest.

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
