# Peer review of "Fibrous Materials for Potential Efficient Energy Recovery at Low-Temperature Heat"

_sustainability, doi:10.3390/su15086567_

Round 1

Reviewer 1 Report

This manuscript is a review article focused on the modification of electrospun fiber to be used as a thermal storage system. This would be very interesting to both academia and industrialists as energy becomes the issue globally to attain sustainability.

This manuscript is well written and some literature work from the past 10 years has been included in this review.   

Nevertheless, the authors should provide a table to compare the advantages, advantages as well as efficiency of all the work which were reviewed. This would be easy to follow for the reader.

Author Response

Please find attached the revision of the manuscript.

Reviewer 2 Report

1. The paper is a good attempt towards review of the elctrospinning along with application of silica aerogel. However, the paper does not describe the silica aerogel in proper terms. Other gels of silica are ineffective thermal insulators.

2. The review lacks the application part. Where are the thermal performance and analysis. There are several articles by Venkataraman et al. or Xiong et al. in this respect.

3. The thermal performance is questionable due to small thickness of the nano/micro fiber layer. There should be sufficient description of the thickness and porosity.

4. What about the structural stability of the aerogel or silica after electrospinning? The emchanicalperformance e.g. compressibility, abrasion resistance etc. should be defined and explained. 

5. A thorough review of literature is necessary to be included.

I recommend a major revision.

Author Response

(The authors gave the same response as above.)

Reviewer 3 Report

The title of the paper is related to material engineering area and how to make fibrous materials but there is no information about their application and efficiency in energy recovery of low-temperature heat in e.g. heat pumps. It corresponds to possibility of use those materials in energy utilization in narrow range. There is no research results how those materials could influence on e.g. heat pumps efficiency or energy consumption.

the main problem of the article is that is focused on making of fibrous materials and their structure but not how they operate in real/research experimental rig. There is no data about how their application will lead to increase low-temperature energy utilization (the cycle efficiency). In my opinion the title and the article should to show those aspects.

Author Response

(The authors gave the same response as above.)

Reviewer 4 Report

This work deals with the topic of electrospinning as a method to recover energy by the use of thermochemical materials. The work is well structured, easily readible and with a clear message. Few minor comments are listed below:

Lines 54-81: a summary of the different methods through the use of a Table/Figure could help readers to focus on the topic.

Lines 139-149: the electrospinning description looks a bit general. Could you highlight more details about this technique? 

Line 161: following what?

Line 213: incomplete square bracket

A final table with pro and cons of the proposed technology is kindly welcomed.

Author Response

(The authors gave the same response as above.)

Round 2

Reviewer 2 Report

The paper is revised.

The language must be improved to reported speech.

Usage of I, we etc. should be avoided.

Example: At the end of Introduction section, " we revised............" should be replaced with " the literature was reviewed......".

Author Response

We thank the reviewer for the suggestion, the text was revised accordingly.